# Physical Activity in Community-Dwelling Older Adults: Which Real-World Accelerometry Measures Are Robust? A Systematic Review

**DOI:** 10.3390/s23177615

**Published:** 2023-09-02

**Authors:** Khalid Abdul Jabbar, Ríona Mc Ardle, Sue Lord, Ngaire Kerse, Silvia Del Din, Ruth Teh

**Affiliations:** 1School of Population Health, Faculty of Medical and Health Sciences, University of Auckland, Auckland 1023, New Zealand; khalid.abdul-jabbar@auckland.ac.nz (K.A.J.); r.teh@auckland.ac.nz (R.T.); 2Translational and Clinical Research Institute, Faculty of Medical Sciences, Newcastle University, Newcastle upon Tyne NE2 4HH, UK; riona.mcardle@newcastle.ac.uk (R.M.A.); silvia.del-din@newcastle.ac.uk (S.D.D.); 3School of Clinical Sciences, Auckland University of Technology, Auckland 1010, New Zealand; sue.lord@aut.ac.nz; 4The Newcastle upon Tyne Hospitals NHS Foundation Trust, National Institute for Health and Care Research (NIHR), Newcastle Biomedical Research Centre (BRC), Newcastle University, Newcastle upon Tyne NE2 4HH, UK

**Keywords:** real-world, validity, reliability, responsiveness, accelerometry, older adults

## Abstract

Measurement of real-world physical activity (PA) data using accelerometry in older adults is informative and clinically relevant, but not without challenges. This review appraises the reliability and validity of accelerometry-based PA measures of older adults collected in real-world conditions. Eight electronic databases were systematically searched, with 13 manuscripts included. Intraclass correlation coefficient (ICC) for *inter-rater reliability* were: walking duration (0.94 to 0.95), lying duration (0.98 to 0.99), sitting duration (0.78 to 0.99) and standing duration (0.98 to 0.99). ICCs for *relative reliability* ranged from 0.24 to 0.82 for step counts and 0.48 to 0.86 for active calories. *Absolute reliability* ranged from 5864 to 10,832 steps and for active calories from 289 to 597 kcal. ICCs for *responsiveness* for step count were 0.02 to 0.41, and for active calories 0.07 to 0.93. *Criterion validity* for step count ranged from 0.83 to 0.98. Percentage of agreement for walking ranged from 63.6% to 94.5%; for lying 35.6% to 100%, sitting 79.2% to 100%, and standing 38.6% to 96.1%. *Construct validity* between step count and criteria for moderate-to-vigorous PA was *r_s_* = 0.68 and 0.72. Inter-rater reliability and criterion validity for walking, lying, sitting and standing duration are established. Criterion validity of step count is also established. Clinicians and researchers may use these measures with a limited degree of confidence. Further work is required to establish these properties and to extend the repertoire of PA measures beyond “volume” counts to include more nuanced outcomes such as intensity of movement and duration of postural transitions.

## 1. Introduction

Physical activity (PA) has been defined as “any bodily movement produced by skeletal muscles that requires energy expenditure” [1]. An increase in PA in older adults is associated with improved muscular strength [2], lower risk of disability [3], and may also protect against cognitive decline [4]. The beneficial effects of PA on functional tasks such as walking in older adults have also been reported [5,6], which is important given that loss of functional ability is associated with functional dependence [7,8] and can lead to social isolation [9] and malnutrition among older adults [10].

Measurement of physical activity in older adults is therefore informative and relevant. The use of body-worn sensors (wearables) to objectively quantify PA is a welcome advancement in the field, given the potential for inaccuracy and bias inherent in self-reported data from questionnaires which are most commonly used [11]. Wearables are defined as devices that can be worn on the skin or attached to clothing to continuously and closely monitor an individual’s activities, without interrupting or limiting the user’s motions (adapted from [12]). Wearables typically incorporate accelerometers to enable continuous (usually seven days), unobtrusive monitoring in real-world environments [13]. Real-world environments generally include the home (which could be retirement villages) and other free-living environments such as parks and cafes, etc. This confers an advantage over data collected in controlled or simulated environments which may have observational bias and other influences [13,14] and are not reflective of real-world conditions [15,16]. Data collected and processed in a controlled laboratory environment, which is usually shorter than those collected in real-world settings, is not reflective of the data collected and processed in real-world environments, especially for older adults [17].

Associated with this advance is the development of novel outcomes from accelerometry data. Frequency counts (e.g., number of sit-to-stand transitions in a day), intensity (e.g., stroll versus run), pattern (accumulation of bouts of walking), and within-person variability give more information than simple volume measures (e.g., total amount of walking time) and therefore provide a greater understanding of the composition of physical activity and the association between it and functional tasks [18].

Several challenges to capturing real-world PA data have been identified. Measurement accuracy in older adults is compromised by the use of walking aids, slower gait, lower level of physical activity intensity, reduced cognitive ability and reduced adherence with research instructions, thus posing technical challenges to detection and capture of movement and analysis [13,19]. Moreover, hardware-related costs and technical competency in dealing with interpretation of accelerometry data, could further add to these challenges [20].

Despite these constraints, there has been a marked increase in accelerometry-based PA research including large-scale, population-based studies (e.g., [21,22]) which in turn has led to issues related to the robustness of these metrics—how reliable, valid and responsive are they? Few studies have examined these questions in-depth (e.g., [23,24]). A recent review update reported that consumer-grade activity trackers tend to overestimate step count (167.6 to 2690.3 steps per day), with slower and impaired gait reducing the level of agreement (<10% at gait speeds of 0.4–0.9 m/s for ankle placement) with reference methods (e.g., ActiGraph) [25]. However, that review included both laboratory and real-world environments, which limits generalizability. Wearables need to be validated under conditions representative of their intended use, that is, in real-world environments [15,16].

In view of the questions that arise from this rapidly expanding field, a synthesis of current research concerning accelerometry-based PA measurement using wearables is required. We posit three key questions: (a) Which PA movements (e.g., sitting, standing, walking) are reliably and validly measured using accelerometers in community-dwelling older adults in real-world conditions? (b) Is the measurement of these PA movements able to show change? (c) How were these PA movements quantified in terms of the type, number, and location of the accelerometers, and duration (time spent) of monitoring? We also report on adherence, usability, and acceptability for wearables where reported.

## 2. Materials and Methods

This systematic review followed the Preferred Reporting Items for Systematic Reviews and Meta-Analyses (PRISMA) guidelines [26] and was registered with the National Health Service PROSPERO database under the registration number: CRD42021228010.

### 2.1. Search Strategy

Systematic searches were conducted across the following eight databases: AMED, CINAHL, Embase, IEEE, Medline, PsycINFO, Web of Science and Scopus. In addition to the above databases, reference lists of review articles and included studies were hand searched to identify additional relevant studies. The search criteria were limited to studies conducted in the English language. An initial search included articles published between 1 January 2010 and 18 January 2021. A follow-up search was conducted on 25 November 2022. A lower limit of 2010 was chosen given the rapid technological advancement in the design and development of accelerometers which are not comparable to those currently used.

The search terms were grouped into four categories and searched in the following sequence: (a) accelerometry/wearable devices (MESH term used—“Fitness Trackers”, “Accelerometry or Actigraphy”, “Wearable Electronic Devices”) (b) physical activity (MESH term used—“exercise”, “running”, “swimming”, “walking”, “motor activity”, “freezing reaction”, “cataleptic”) (c) older adults (MESH term used—“aged”, “aged, 65 and over”) (d) clinimetric properties (MESH term used—“reproducibility of results”, “sensitivity and specificity”). Further details of the search strategy are provided in Appendix A.

### 2.2. Inclusion and Exclusion Criteria

Table 1 shows the eligibility criteria that were employed in this review.

### 2.3. Data Extraction and Abstraction

All searches were imported and screened for duplicates in EndNote X9 (Version 3.3). The titles were initially screened by KAJ in EndNote X9, then the selected titles and their respective abstracts were exported as a CSV file and imported into a web-based systematic review software—Rayyan [27]. Thereafter, the remaining abstracts were screened by two reviewers (RMA & KAJ) in a blinded process. Disagreements over inclusion were adjudicated and resolved by a third reviewer (SL). Reasons for exclusion were recorded for abstracts based on the inclusion/exclusion criteria. Following the abstract screening, the remaining full texts were independently reviewed by two reviewers (RMA & KAJ).

A data extraction form was used to standardize the information extracted from each article. KAJ extracted the data which were verified by RMA.

### 2.4. Clinimetric Properties

*Inter-rater reliability* was established as the degree of agreement between two independent observers of duration of activities from videos and reported as intra-class correlation (ICC, 95% CI). *Relative reliability* was established as the degree of agreement in terms of ranks or position of individuals within a group and reported as intra-class correlation (ICC, 95% CI). *Absolute reliability* was established as the degree of agreement in terms of precision of the individual measurements and reported as minimal detectable changes (that was calculated using standard error of measurement). *Responsiveness* was established as the ratio between minimally clinically important change on the measure and mean squared error of the response obtained from an analysis of variance model and reported as Guyatt’s responsiveness (GR) coefficient [28].

*Criterion validity* was established either as agreement between a gold standard reference and accelerometry or as percentage of agreement between video observation and accelerometry and was reported as ICC or as F-Score (for comparison between different algorithms) or as sensitivity, specificity, accuracy, precision, or positive predictive values.

Bland–Altman plots [28,29,30,31] or modified Bland–Altman plots [32,33] described limits of agreement and systemic errors. *Construct validity* was tested between step counts and moderate-to-vigorous PA and was reported as correlation, *r_s_* (Spearman’s Rho).

### 2.5. Risk of Bias Assessment

The Appraisal tool for Cross-Sectional Studies (AXIS) checklist was used to evaluate the risk of bias for all studies included in this review [34]. Two reviewers (RMA and KAJ) independently assessed the quality of the studies, with a third reviewer (SL) settling any disagreements.

## 3. Results

### 3.1. Study Selection

The initial search identified 13,872 records, of which 6206 duplicates were removed. The remaining 7666 titles were screened, resulting in 768 records carried through to abstract review. The updated search conducted on the 25th of November 2022 identified 3179 records, of which 1455 duplicates were removed. The remaining 1724 titles were screened by KAJ resulting in 144 records for the abstract stage (Figure 1). Two reviewers (RMA and KAJ) screened the abstracts based on the inclusion and exclusion criteria and identified 79 records for full-text review. Thirteen records passed through to the final full-text review stage. Two further publications were retrieved from reference lists, one of which was classified as a Research Letter. Reasons for exclusion included study setting other than “real-world” such as a semi-structured or simulated real-world environment (*n* = 24); PA metrics relevant to the review were not reported (*n* = 16); or the study did not report any clinimetric data (*n* = 12). EndNote was used to index all records.

### 3.2. Quality of Studies

Thirteen studies included in this review achieved a minimum score of 70% (i.e., 12 out of a possible 17) based on the AXIS toolkit (Table 2). Thus, two studies were excluded from the review due to methodological weaknesses [35,36].

**Table 2 sensors-23-07615-t002:** Methodological quality assessment of selected articles based on AXIS checklist [34].

Ref.	Q1	Q2	Q3	Q4	Q5	Q6	Q7 ^1^	Q8	Q9	Q10	Q11	Q12	Q13 ^1^	Q14 ^1^	Q15	Q16	Q17	Q18	Q19 ^2^	Q20	Total Score
[37]	1	1	0	0	1	0	-	1	0	1	1	1	-	-	1	1	1	1	1	1	13
[36]	1	1	0	0	0	0	-	0	1	1	0	1	-	-	1	1	1	1	1	1	11
[38]	1	1	0	0	0	0	-	1	1	1	1	1	-	-	1	1	1	1	1	1	13
[31]	1	1	0	0	1	0	-	1	1	1	1	1	-	-	1	0	1	1	1	1	13
[39]	1	1	0	0	0	0	-	1	1	1	1	1	-	-	1	1	1	1	1	1	13
[32]	1	1	0	0	1	0	-	1	1	1	1	1	-	-	1	1	1	1	1	1	14
[40]	1	1	0	0	0	0	-	1	1	1	0	1	-	-	1	1	1	1	1	1	12
[29]	1	1	1	1	0	0	-	1	1	1	1	1	-	-	1	1	1	1	1	1	15
[41]	1	1	0	1	0	0	-	1	1	1	1	1	-	-	1	1	1	1	1	1	14
[28]	1	1	1	1	0	0	-	1	1	1	1	1	-	-	1	1	1	1	1	1	15
[30]	1	1	0	1	1	0	-	1	1	1	1	1	-	-	1	1	1	0	0	1	13
[42]	1	1	0	0	1	0	-	1	1	1	1	1	-	-	1	1	1	0	1	1	13
[35]	1	1	0	1	1	0	-	0	0	0	0	0	-	-	1	1	1	1	1	1	10
[33]	1	1	0	1	1	1	-	1	1	0	0	1	-	-	1	1	1	1	1	1	14
[43]	1	1	0	1	1	0	-	1	1	0	0	1	-	-	1	1	1	1	1	1	13

Note: “Q” refers to question. So “Q1” implies “Question 1”. Each 1’s, which are in green fonts, represents an affirmative appraisal score for that question, while each 0’s, which are in red fonts, represents a negative appraisal score for that question. Please see Downes et al. [34] for more details. ^1^ These questions related to non-responders were not included in the tabulation of the scores. ^2^ A negative response to this question “*Were there any funding sources or conflicts of interest that may affect the authors’ interpretation of the results?*” is taken as a score of “1” and vice versa.

### 3.3. Characteristics of the Studies

All studies bar one recruited a Caucasian population [43]. The mean age of the total sample was 74.9 ± 6.1 years, and 62.4 ± 20.2% were females. Participants over 80 years of age who were included (*n* = 45), were mainly frail but ambulant [32,33,41,43]. Participants were recruited either through ongoing studies or advertisement (letter, flyer, word-of-mouth, etc.) or through convenience sampling from senior citizen centers. The sample sizes ranged from 5 to 50 participants. All 13 studies used wearable sensors incorporating a tri-axial accelerometer with an average duration of 108.2 ± 128.8 h of data. Four studies investigated duration of physical activities such as sitting, standing, walking and lying [33,37,40,41], three investigated only walking (gait) bouts [32,38,42], and six studies focused mainly on step counts as their key outcome measures [28,29,30,31,39,43]. Two studies used only proprietary accelerometers [32,41], one included both proprietary and commercially available accelerometers [37], and ten studies used commercially available accelerometers. Four studies reported *inter-rater reliability* (*n* = 63) [33,37,40,41] and one study reported *relative reliability*, *absolute reliability* and *responsiveness* (*n* = 50) [28]. Twelve studies reported *criterion validity* (*n* = 321) with one study reporting *construct validity* (*n* = 30) [39]. All 13 studies (total *n* = 351) were cross-sectional validation studies and were from Australia (*n* = 62) [30,39], Israel (*n* = 12) [38], Japan (*n* = 44) [43], The Netherlands (*n* = 25) [40,41], New Zealand (*n* = 38) [32,33], Norway (*n* = 16) [37], Slovenia (*n* = 50) [28], Switzerland (*n* = 37) [42], UK (*n* = 25) [29] and USA (*n* = 35) [31] (Table 3 and Table 4).

### 3.4. Study Protocol

Study protocols for validity testing varied with respect to the reference standard, the outcome of interest, environment and duration of testing, as well as the location of sensors.

### 3.5. Reference Standard

Six studies validated consumer-grade wearables with research-grade reference accelerometers [28,29,31,38,39,42]. Five of the studies used the video/visual method as their “gold standard” for their validation reference [32,33,37,40,41]. One study used both video as well as research-grade reference accelerometers [30]. One study used the doubly labelled water (DLW) method as their reference for their validation [43] (Table 5). The DLW method is an established technique for measuring energy expenditure. This method is based on the estimation of the rate of CO_2_ elimination from the body [46]. 

### 3.6. Outcomes

Step count was reported as the main outcome in six studies [28,29,30,31,39,43]. Three studies focused on gait bouts [32,38,42], whilst the duration of walking, sitting, standing and lying was reported in the remaining four studies [33,37,40,41] (Table 5).

### 3.7. Environment

Nine studies collected and validated real-world accelerometry data exclusively within the participants’ home/retirement village environment [29,30,31,32,37,38,41,42,43]. One study investigated criterion validity in a controlled setting within a retirement village as well as in participants’ home environment [33]. Dijkstra and colleagues investigated criterion validity in the laboratory environment with 20 participants and also carried out further validation in a real-world home environment with five participants [40]. Burton and colleagues investigated intra-rater reliability using the two-minute-walk test (2MWT) in a laboratory environment, but construct validity in the home [39]. Kastelic and colleagues conducted a test battery that included common real-life tasks, within the laboratory environment as well as an uncontrolled free-living study [28]. For the purposes of this review, we have included only the home or free-living environment data in our analysis.

### 3.8. Duration of Wear

Duration of wear was mixed among the studies and ranged from 14 days duration to under 10 min: <10 min (*n* = 45) [32,33]; 30 min (*n* = 25) [40,41]; 100 min (*n* = 16) [37]; 12 h (*n* = 37) [42]; two days (*n* = 35) [31]; four days (*n* = 50) [28]; seven days (*n* = 57) [29,30]; ten days (*n* = 12) [31] and 14 days (*n* = 74) [39,43]. None of the studies exceed the 14-day duration. Duration of wear was influenced by the choice of criterion in the sense that studies relying on research-grade accelerometers [28,29,30,31,39,42] and DLW method [43] as their reference standard captured un-instructed daily activities (excluding water activities) during waking hours and exceeded a 12 h period. By contrast, studies that employed video or direct observation as reference [32,33,37,40,41] limited their duration of real-world observation to a maximum of 100 min, with two studies capturing less than 10 min of activities [32,33] (Appendix A).

### 3.9. Sensor Location

The most common location of wear was the wrist (*n* = 205) [28,29,31,37,39,42] followed by the back (lower back or back of the waist) (*n* = 104) [32,33,40,43]. One study also extended the validation for the Misfit Shine accelerometer to the waist as an additional location of wear because this device was designed to be worn in both places [29]. One study used the wearable on the right hip (*n* = 32) [30] and another used the wearable as a necklace (*n* = 20) [41]. Awais and colleagues investigated data from participants who wore four accelerometers concurrently—on the wrist, chest, lower back and thigh (*n* = 16) [37] (Appendix A).

The location of sensors appeared to influence accuracy. Studies that use a single sensor close to the participant’s center of gravity such as the waist [29], hip [30] or lower back [32,33,40] reported higher sensitivities than those placed on the wrist or around the neck, supporting the findings of a recent review [25].

### 3.10. Reliability

*Inter-rater reliability* conducted within real-world conditions was reported in four studies (*n* = 63) [33,37,40,41]. Dijkstra et al. [40] reported inter-rater reliability of activity durations of ten participants by two independent observers (via video analysis)—walking (0.95), sitting (0.78), standing (0.99) and lying (0.98). Taylor et al. [33] also reported excellent inter-rater reliability between two independent observers on ten randomly selected video footage—walking (0.94), sitting (0.99), standing (0.98) and lying (0.99). Geraedts et al. [41] reported ICC for inter-rater agreement of the video observation was 0.91 in the free movement protocol. Awais et al. [37] reported that the overall level of agreement of out-of-lab activities was above 0.90 for one randomly selected video that was chosen to be rated by five independent raters.

*Intra-rater reliability* conducted within real-world conditions was reported as *relative* and *absolute reliability*. *Relative reliability* of step counts from commercial accelerometers ranged from poor to good for both single-day averages as well as three-day averages. The results were similar for average active calories (which was based on the differences between total calories computed by the accelerometers and estimated basal metabolic rate based on Harris and Benedict [51]). *Absolute reliability* was generally better (i.e., lower) for averaged measures—step count and active calories—of three days compared to single-day measures [28].

### 3.11. Validity of Accelerometers

The overall sample size for testing criterion validity was *n* = 321 including diverse populations and incorporating a range of study protocols. *Criterion validity* between research-grade wearables and consumer-grade wearables was excellent for step counts measured at the right hip: ICC = 0.94 (95%CI [0.88, 0.97]) (FitBit One/Zip versus ActiGraph GT3X+) [30] and the waist: ICC = 0.96 (95%CI [0.91, 0.99]) (Misfit Shine versus ActiGraph GT3X+) [29] and ICC = 0.91 (95%CI [0.79, 0.97]) (NL2000i) [29], but lower on the wrist: ICC ranged from ICC = 0.83 (95%CI [0.59, 0.93]) (Misfit Shine versus NL2000i) to ICC = 0.86 (95%CI [0.67, 0.94]) (Misfit Shine versus ActiGraph GT3X+) [29]. The average daily step count between consumer-grade wearables and reference devices was overestimated in two studies [28,30]. Results were mixed in another, which employed two different locations of wear (wrist and waist) as well as two different research-grade reference devices (ActiGraph and NL2000i) [29]. Garmin Vivosport and Garmin Vivoactive 4s performed much better than Polar Vantage M for step counts—0.98 versus 0.37 and 0.95 versus 0.37, respectively [28]. Briggs et al. [31] found no significant difference (*p* = 0.22) between the daily step count from wrist-worn Garmin Vivosmart HR and the reference, hip-worn ActiGraph GTX3X+: ICC = 0.94 (95%CI [0.88, 0.97]). This study also reported that the differences due to step counts derived MVPA were reduced using age-specific cut-offs [31]. Kastelic et al. [28] reported that computed measures such as activity calories, which were derived from accelerometry and heart rate data, did not perform as well as step counts from the accelerometry. The agreement between activity calories for all three devices was lower than the agreement between steps counts: Garmin Vivosport—0.58, Garmin Vivoactive 4s—0.55 and Polar Vantage M—0.15. Two studies validated algorithms developed to detect the duration of gait bouts estimated from a single wrist worn consumer-grade wearable. Brand et al. collected ten days of data from 12 older adults, and reported that the new algorithm had 76.2% accuracy, 29.9% precision, 67.6% sensitivity and 78.1% specificity for detecting gait bouts. Soltani et al. collected 12 h of data from 37 older adults, and reported even better scores—accuracy was 95.2%, precision was 71.8%, sensitivity was 87.1% and specificity was 96.7% for detecting gait bouts. They also compared their method with previously published algorithms. Their algorithm’s F1-score was 74.9%, which was better or close to earlier studies which utilized multiple accelerometers. [42,47].

Studies that used the observer/visual/video method to validate their data with accelerometer-based algorithms mainly reported their results using sensitivity, specificity and positive predictive values. Chigateri et al. [32] reported agreement between uSense algorithm classification compared to video labelling (frame-by-frame analysis) for walking and non-walking during unscripted activities (real-world) as 88.7% (74.9–96.4) and 92.2% (89.5–95.7), respectively; however, the algorithm systematically overestimated walking behaviour. The mean difference between the algorithm and video categorization was 26.5 s. Dijkstra et al. [40] reported sensitivity for walking—93.5%, lying—98.7%, sitting—83.2% and standing—80.1%, between the output of DynaPort MoveMonitor and video analysis values (durations). Similarly, Taylor et al. [33] reported sensitivity of walking (locomotion)—92.2%, lying—100%, sitting—94.5% and standing—38.6% when comparing the output from DynaPort MoveMonitor and the video analysis values (durations). Median absolute percentage error was reported as: walking, 0.2% (inter quartile range (IQR), −4.3% to 14.0%); lying, 0.3% (IQR, −4.2% to 21.4%); sitting, −22.3% (IQR, −62.8% to 10.7%); and standing, 24.7% (IQR, −7.3% to 39.6%). The authors noted that 45.6% of the unscripted standing time was misclassified as sitting and 5.3% of the unscripted sitting time as standing [33]. Geraedts et al. [41], who used the accelerometer as a necklace instead of attaching it to the lower back [32,33,40], reported lower sensitivities: walking—63.6%, lying—35.6%, sitting—79.2% and standing—61.3%. One study compared the classical machine learning-based physical activity classification algorithms and deep-learning based physical activity classification algorithms with previous studies in detecting various activities [37]. The F-Score for: walking—94.5%, lying—98.5%, sitting—99.9% and standing—96.1%. The F-score computed based on various sensor configurations (lower back; wrist; thigh; chest; lower back and thigh; lower back, chest, and thigh; lower back, wrist, chest, and thigh) for sitting [range: 99.7% to 100.0%], lying [84.7% to 98.5%], standing [91.6% to 96.1%] and walking [86.9% to 94.5%] activities. Combining more sensors produced better scores.

*Construct validity* of step counts and moderate-to-vigorous physical activity (MPVA) between GENEactiv accelerometer and consumer-grade wearables—Fitbit Flex and Fitbit ChargeHR—was reported as a moderate level of agreement between the devices (ICC*_Flex_*: 0.68; ICC*_ChargeHR_*: 0.72) [39].

In summary, these results suggest that step counts and duration of walking, lying, sitting and standing can be measured robustly to a certain degree using a single accelerometer. However, further work is required to understand better how the location of wear and type of reference standard affect accuracy.

One study [43] investigated the validity of a triaxial accelerometer against the doubly labelled water method (DLW) for total energy expenditure reported that the 24 h average metabolic equivalent (MET) of Actimarker was significantly correlated with the PA level assessed by DLW but significantly underestimated it (*p* < 0.001). Furthermore, the correlation between daily step counts and PA level of DLW was moderate: *R*^2^ = 0.248 (*p* < 0.001).

### 3.12. Responsiveness of Accelerometers

Only one study reported on the responsiveness of accelerometry (i.e., the capacity of an accelerometer to identify possible changes in PA outcomes associated with a clinical condition over time) [28]. Single day measure of step counts performed better than average three-day measures for Garmin Vivoactive 4s (GR—0.411 vs. 0.041) and Polar Vantage M (0.126 vs. 0.060), but not for Garmin Vivosport (0.022 vs. 0.288). However, all three devices showed relatively weak to moderate responsiveness for active calories (GR > 0.232) for both single-day as well as averaged-day measurements, except for Garmin Vivosport (Single day GR = 0.073) [28] (Table 5).

### 3.13. Acceptability and Adherence of Accelerometers

Only one study planned and purposefully measured adherence. Geraedts et al. [41] reported 100% adherence during daytime and 80% during sleep from a necklace sensor worn for seven days. The authors also collected information on the level of comfort, weight, size and usability of the sensor when worn during the daytime using a user-evaluation questionnaire on a scale of 1 to 5. They reported a high mean score of 4.4 ± 0.6 and concluded that user acceptance was high. Three studies reported adherence based on missing data [28,29,39]. Farina et al. [29] required participants to wear five devices (two on the wrist and three on the waist) over seven consecutive days and reported excluding three participants (12%) from their analysis due to receiving less than four days of data from the reference device, which indicated that adherence was low for longer durations of data capture. Burton et al. [39] reported that close to 50% of participants had some missing data from their wrist-based wearables over 14 days, also suggesting declining levels in adherence with increasing duration of data capture. Kastelic and colleagues reported the adherence of wearing three different accelerometers on the non-dominant wrist (together with a reference accelerometer on the waist) over 12 days, each device for four days, based on wear time. The wear time compliance with the Polar Vantage M, Garmin Vivoactive 4s and Garmin Vivosport was as high as 24.0 ± 0.1 h/day, 23.9 ± 0.5 h/day and 23.9 ± 0.5 h/day, respectively. None of the four studies reported age- or gender-related differences (Appendix A).

### 3.14. Summary of Results

Table 6. summarizes the clinimetric properties of accelerometry-based PA measures of older adults collected in real-world conditions.

## 4. Discussion

This review is the first to our knowledge to examine the reliability and validity of accelerometry-based PA measures of older adults collected in real-world conditions. Moderate to strong ICCs for inter-rater reliability and criterion validity tentatively establish step count, duration of walking, sitting, standing and lying as robust outcomes. Variations in the methods such as location of sensors and duration of wear highlight differences in the strength of the validity and reliability of the outcome measures. This also points to a need for standardization of protocols of wearing accelerometers in future studies. However, this review identified limitations in the current literature, specifically that most of the outcomes are limited to volume metrics.

### 4.1. Reliability of PA Measures

Good to excellent inter-rater reliability was observed for the durations of sitting, standing, walking and lying activities. Inter-rater reliability of step counts in real-world environment was not reported. Intra-rater reliability differed by the brand and the type of measures. The Garmin Vivosport and Garmin Vivoactive 4s had better relative and absolute reliability than the Polar Vantage M for both step counts as well as active calories. Derived metrics from step counts, such as activity intensities (e.g., MVPA), were not as reliable as steps counts. None of the studies investigating duration of PA activities reported intra-rater reliability. The reasons for omitting reliability testing were not discussed by the authors. This omission limits our understanding as to whether accelerometry-based PA measures such the durations of sitting, standing, walking and lying activities are affected by the individual observers, when captured in real-world conditions.

### 4.2. Validity of PA Measures

The most common “gold-standard” reference for criterion validity was using research-grade accelerometers, followed by the use of video or direct observation. In all but one reported study, a single tri-axial accelerometer was sufficient to distinguish PA validly. However, there was a lack of homogeneity for real-world assessments with respect to sensor location and duration, the brand of accelerometer employed, and the instructions given to participants when carrying out *uninstructed* daily activities.

As noted above, the duration of wear varied amongst studies which is partially attributable to the level of intrusiveness of the reference devices used. There seems to be no consensus on the minimum length of duration for accelerometry-related validation studies, but a minimum of 30 min of semi-structured activities has been previously recommended for real-world settings [52]. Capturing, processing and annotating videos that are several days in length might be challenging, and the alternative seems to be to aim to capture as many commonly performed activities within a shorter timeframe [52]. Additionally, merging and synchronizing of data is challenging, although the use of platforms seems to offer some promise [53]. Intrinsic factors (motivation, personal preferences) and extrinsic factors (weather, environment) may affect habitual physical activity performance [15,54]. Although this seems to be a reasonable compromise between duration and practicality, it is questionable as to whether the variations in intrinsic and extrinsic factors within daily PA could be captured within such a timeframe.

Chigateri et al. [32] and Dijkstra et al. [40] provided limited instructions for unstructured real-world activities, e.g., “what they normally do during the day”, whereas others were more explicit. Taylor et al. [33] informed their participants to include common activities such as walking, lying, sitting, and standing, while Geraedts et al. [41] included common household chores such as vacuuming and clearing dishes. It is noteworthy that among these commonly reported four PA—walking, lying, sitting and, standing—the sensitivity for sitting and standing were relatively lower than the former two. The use of a single sensor on the lower back was not able to sufficiently distinguish sitting from standing, which could have misclassified these two activities in two studies [33,40]. However, Chigateri et al. [32] reported that walking duration was overestimated with the uSense wearable device and postulated that there was a higher likelihood for algorithms to overestimate walking duration since inactive durations such as ‘pauses during walking’ between walks could have been misclassified as walking time [32]. Awais et al. [37] dataset consisted of 15 common free-livings activities (see [44,45]) that were performed in an order that suited the participants’ preferences, but with no other instructions. This study compared the use of machine learning and deep-learning techniques to classify data from four accelerometers, concurrently worn on four different locations on the body, as walking, lying, sitting, and standing activities. Although the use of additional accelerometers to classify activities produced much better results than studies that used a single sensor, it was not conclusive as to which technique—machine learning versus deep-learning technique—was superior, since the results of both methods plateaued [37].

Steps counts were generally overestimated by commercial-grade wearables, but the evidence was not conclusive since different brands of accelerometers elicited different results [28]. Although step count derived metrics generally did not perform as well as direct step counts, and the choice of age-specific cut-offs could improve the accuracy [31].

### 4.3. Study Protocol

Validity and accuracy of the metrics varied with the duration of data collection. Soltani et al. [42] achieved very high accuracy in identifying gait bouts from 12 h of data. Brand et al. [38], who also used the wrist but collected data up to ten days, reported worse results. However, both studies used different accelerometers, and the choice and location of wear of their references was also different—one used the Axivity AX3 on the lower back, while the other used the ActiGraph GT9X Link on the shank. Furthermore, the algorithms implemented by these two studies were also different [31,42].

These discrepancies highlight the need for standardization of the methodology used in validation studies to allow comparison between their results and findings. Future validation studies should aim to adopt recommended methods and protocols relevant for community-dwelling older adults [45,52].

Interestingly, only one study investigated whether wearables could detect change over time, but the findings were mixed, inconclusive and device-specific [28]. The responsiveness of single-day measures of step counts was generally better than the three-day average, but this needs to be cautiously interpreted. The lack of evidence on responsiveness from more studies may reflect the recruitment of generally healthier older adults. There is greater impetus to establish responsiveness for people with neuro-musculoskeletal conditions, for example those with age-related degenerative conditions such as osteoarthritis [55].

### 4.4. Adherence to Study Protocol

The duration and location of wear of the sensor affected the level of adherence. Wrist-worn sensors yielded a high level of adherence, but increasing the duration of data capture could reduce the level of adherence and compliance [28,29,39]. Although wearing sensors on the wrist may be more natural than other locations such as the lower back and the ankle, there was a possibility that older adults might forget to put them back on after they had removed them, perhaps during sleep. There was a high level of adherence in wearing the sensors as a necklace, but at the expense of sensitivity, perhaps because there was no need to remove them during sleep and studies constituted a high proportion of females who may already be in the habit of wearing necklaces. Only one study investigated the level of acceptance of the wearables they tested, possibly because the investigators were developing a new wearable prototype [41].

Real-world validation studies of older adults for different intensities of PA, such as different speed or intensity of walking, are missing in the literature. We know that the accuracy of step counts was lower in participants who walked with a slower gait speed [25] or walked with lower intensity [56]. Also lacking are validation studies that test more nuanced metrics such as the duration of postural transition, including sit-to-stand and stand-to-sit, which are important indicators of functional mobility and lower limb strength. Real-world postural transitions, similar to other PA, are ecologically more valid when performed at home as they are executed in a familiar environment [52].

Despite this, there is a growing body of inference-based evidence from studies that use accelerometry to investigate associations between mortality, health and functioning in large populations. These studies indirectly examine aspects of validation such as construct validity [57] and predictive validity [58], thereby providing some assurances.

### 4.5. Strengths and Limitations of the Review

This systematic review used a comprehensive search strategy of eight databases, included clear inclusion and exclusion criteria, utilized the AXIS checklist to access risk of bias, and followed the PRISMA guidelines. It also adopted the blinded adjudication process for the abstract and full-text review. The process followed in the review was designed to minimize bias and increase the transparency of the reporting.

Limitations included a focus on studies in the English language and exclusion of grey literature. Secondly, the sample size for most of the studies was small and predominantly female. Finally, not all the studies reported on the reliability of the wearables, and of those that did, all failed to report test–retest reliability. Both these latter limitations could have weakened the overall strength of the studies reported. In addition, larger-scale and longer-duration studies could better inform us on the level of adherence in wearing the accelerometers among older adults.

## 5. Conclusions and Implications for Future Research

Step counts, duration of walking, sitting, standing and lying are reliably and validly measured using accelerometers in community-dwelling older adults in real-world conditions. However, only step counts have been reported to show change over time.

Robust outcomes from accelerometry monitoring of PA are limited to ‘volume’ counts such as number of steps and duration of sitting, standing, walking and lying, which points to the need for further research on nuanced PA outcomes to provide more in-depth understanding on how PA affects functional tasks. Wrist-worn and neck-worn accelerometers are not as metrically robust as those worn at the waist, hip and lower back. Adherence and usability are negatively associated with duration of wear.

To extend the field of research, more real-world studies are needed, in particular, more studies that focus on generally healthy older adults, investigating more nuanced aspects of PA such as intensity of movement (e.g., slow walk versus running) and duration of postural transitions. Data from non-Caucasian populations are also needed. More longitudinal studies are needed to investigate the responsiveness of the metrics, for example, whether step counts are sensitive to detect fall risk in healthy community-dwelling older adults. Finally, future studies should also investigate wearability and acceptance of their wearables in larger sample cohorts. This will inform researchers on whether such wearables could be used in longer-term data collection processes.

## Figures and Tables

**Figure 1 sensors-23-07615-f001:**
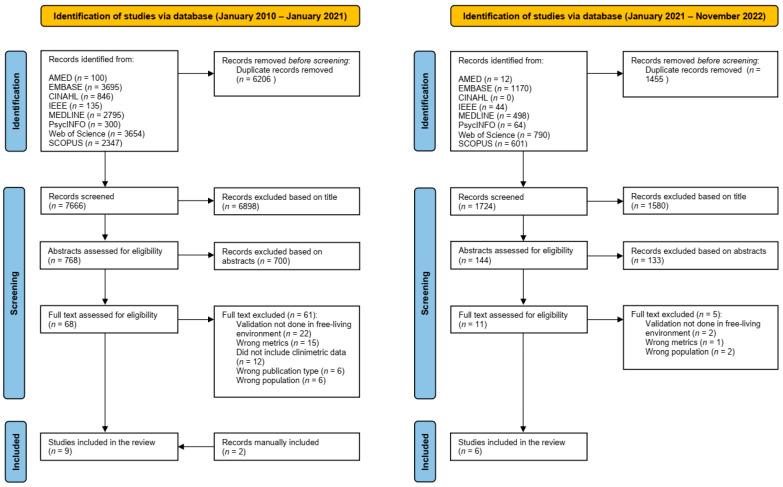
PRISMA flow chart of study design.

**Table 1 sensors-23-07615-t001:** Eligibility criteria for article selection in this review.

Factors	Inclusion Criteria	Exclusion Criteria
Language	published in the English language	published in any other than English
Time frame	1 January 2010 to 25 November 2022	not applicable
Setting	living in the community and including those in retirement villages	studies only including participants from aged residential care settings, including supported living, assisted living, nursing homes, care homes
Topic	data were collected in a real-world environment; studies that reported at least one clinimetric property—reliability, validity, responsiveness	studies in which PA were not objectively quantified using accelerometry; studies not concerned with PA metrics; studies that only reported only qualitative data; studies that only investigated disease-specific populations, e.g., Parkinson’s Disease, Alzheimer’s Disease, stroke; studies that only include laboratory-based measurement
Population	aged 65 years and above	children; adults < 65 years
Publication type	peer-reviewed publications; randomized controlled studies, cross-sectional and longitudinal (retrospective and prospective) quantitative studies, case–control studies	conference abstracts; posters; study protocols; reviews; meta-analyses; grey literature

**Table 3 sensors-23-07615-t003:** Sample size and basic demographic information of each population from the studies included in the systematic review.

Study	Sample Size (*n*)	Mean Age (yrs) ± SD	Female (%)
Awais et al., 2021, Norway [37]	20 *	76.4 ± 5.6	75.0
Brand et al., 2022, Israel [38]	12	76.1 ± 7.3	75.0
Briggs et al., 2022, US [31]	35	73.7 ± 6.3	6.0
Burton et al., 2018, Australia [39]	30	74.20 ± 5.78	64.5
Chigateri et al., 2018, New Zealand [32]	23	80.5, range: 75–92	74.0
Dijkstra et al., 2010, Netherlands [40]	5	73.0 ± 6.2	80.0
Farina et al., 2018, UK [29]	25	72.5 ± 4.9	48.0
Geraedts et al., 2015, Netherlands [41]	Frail (*n* = 7); Non-frail (*n* = 13)	Frail (84.1 ± 2.6); Non-frail (77.3 ± 5.5)	Frail (100%); Non-frail (61.5%)
Kastelic et al., 2021, Slovenia [28]	16 (Polar Vantage M); 17 (Garmin Vivoactive 4s); 17 (Garmin Vivosport);	74.0 ± 5.0	56.3 (Polar Vantage M); 70.6 (Garmin Vivoactive 4s); 58.8 (Garmin Vivosport);
Paul et al., 2015, Australia [30]	32	67.7 ± 5.7	63.0
Soltani et al., 2020, Switzerland [42]	37	64.0 ± 11.0	51.3
Taylor et al., 2014, New Zealand [33]	Independent (15); Long-term care (7)	Independent (87.9 ± 5.1); Long-term care (88.3 ± 5.3)	Independent (78.6%); Long-term care (85.7%)
Yamada et al., 2018, Japan [43]	With sporting habits (9); Without sporting habits (19); Frail (16)	With sporting habits (71 ± 5); Without sporting habits (74 ± 7); Frail (84 ± 8))	With sporting habits (33.3%); Without sporting habits (68.4%); Frail (50%)

* Only 16 were analysed.

**Table 4 sensors-23-07615-t004:** Design, settings and aims of studies included in the systematic review.

Study	Design and Settings	Aims	Inclusion Criteria	Exclusion Criteria	Strengths of Study	Limitations of Study
Awais et al. [37]	Cross-sectional—validation study; dataset (ADAPT) used in the study was based on Bourke et al. ^1^ All free-living activities were performed in participants’ homes.	To use deep learning approach to classify physical activity, and to compare the performance of the deep learning approach with classical machine learning approach. Algorithms tested were PAC-LSTM, PAC-All-Feat, PAC-CFS, PAC-FCBF and PAC-ReliefF.	Aged 65 years and over; able to walk 100 m without walking aids; accepts oral instructions; living independently.	ND *	Study included a well annotated dataset of older people in free-living conditions.	Dataset was not large for machine learning and deep learning type of studies.
Brand et al. [38]	Cross-sectional—validation study; free-living measurements were carried out in participants’ homes and in the community.	To detect gait from wrist worn tri-axial accelerometer recordings of daily living of older adults, using an anomaly detection algorithm and compared its performance to four previously published gait detection algorithms.	ND *	ND *	ND *	The current study did not investigate shorter gait bouts (<6 s).
Briggs et al. [31]	Cross-sectional—validation study; free-living measurements were carried out in participants’ homes and in the community.	To determine the content validity of the Garmin Vivosmart HR compared to ActiGraph GT3X+ for the domains of daily step count and MVPA ^2^.	Veterans aged ≥65; able to perform ADL; able to follow instructions in a group setting; free from ischemic heart and severe lung diseases. Does not require walking assists devices; able to provide written consent.	ND *	ND *	The participants were mainly male—this limits the generalizability of the findings.
Burton et al. [39]	Cross-sectional—validation study; laboratory validation was carried out in outside research lab with walking space, e.g., hallway. Free-living measurements were carried out in participants’ homes and in the community.	To test the reliability and validity of two fitness trackers (Fitbit Flex and Fitbit ChargeHR) by step count when worn by older community-dwelling people.	Aged 65 years and over; living in Perth; owns a smart phone or tablet; understands English and no medical condition which made participation in the study unsafe (i.e., must be able to walk for a minimum of 2 min unassisted).	ND *	First study investigating an older population in both laboratory and free-living (over 14 days) environments using the Fitbit Flex and Fitbit ChargeHR.	Reference device (GENEactiv) did not quantify exact parameter (step count) as devices being validated. Limited generalizability as participants were older with good mobility.
Chigateri et al. [32]	Cross-sectional—validation study; scripted and unscripted task were performed in participants’ homes.	To validate the performance of uSense in detecting non-sedentary activities, differentiating walking and non-walking episodes for frail older people aged 75 years and above in free-living environment.	Aged ≥75; the ability to walk independently with or without a walking aid for a minimum of 20 m.	Any significant medical, orthopaedic or neurological conditions that would contraindicate normal activity, e.g., acute inflammatory arthritis, pneumonia, unstable or acute heart failure, allergy to surgical adhesive tape.	ND *	Waist-worn sensor unable to discriminate sitting and standing postures. Shuffling from walking were not explicitly differentiated.
Dijkstra et al. [40]	Cross-sectional—data collected at participants’ homes.	To determine the accuracy of the DynaPort system for detecting gait (walking, shuffling) and postures (lying, sitting, standing) in community-dwelling older adults during activity sequences in a home environment.	ND *	Impairments or diseases (e.g., orthopaedic, neurological) that could affect the performance of daily activities such as walking, getting in and out of a chair or bed.	ND *	Study was based on large number of fairly short activities. Results may further improve during prolonged monitoring when older adults are expected to have longer periods of sedentary activity.
Farina et al. [29]	Cross-sectional—validation study; validation was carried out in the research lab. Free-living measurements were carried out in participants’ homes and in the community.	To validate two commercially available devices (Misfit Shine and Fitbit Charge HR) against two well-validated research-grade, tri-axial activity monitors (ActiGraph GT3X+ and New Lifestyle NL2000i) in community-dwelling older adults in free-living conditions. To determine whether the Misfit Shine, which is designed to be worn on the wrist or waist, is valid to wear on one or both locations.	Community-dwelling older adults; aged 65–84	Not independently ambulatory or use of a walking aid (self-reported).	ND *	The accuracy of the consumer-level devices is based upon the agreement with existing reference devices that assumes validity in an older population. The population was composed solely of healthy older adults who were independently ambulatory, thus might not be generalizable on frail older populations. Did not have objective means to determine whether participants wore the device in accordance with the protocol.
Geraedts et al. [41]	Cross-sectional—validation study; all data collected at participants’ homes.	To assess the validity of a sensor-based method to detect time-on-legs (standing) and daily life mobility related postures in older adults based on a necklace-worn motion sensor. To evaluate user opinion about the practical use of the sensor.	Community-dwelling or living in an older adult home; aged ≥70 years; able to walk 10 m without support or with a cane or walker.	Orthopaedic impairments that debilitate the ability to walk unsupported for ten metres; total hip or knee replacement surgery in the previous six months; having had a stroke within the last six months; Parkinson’s disease stage 4/5 or other neurologic diseases that can impair daily functioning or visual problems to a degree that make it impossible for the participant to accurately read the questionnaires or walk around safely.	Validation was carried out in semi-structured home environment and not in lab settings. Both frail and non-frail participants were included in the study.	Free living data collection was limited to 30 min only. Outdoor activities, such as cycling was not included in study. Participants performed movements in a rushed manner to complete several tasks which has an impact on accuracy.
Kastelic et al. [28]	Cross-sectional—validation study; free-living measurements were carried out in participants’ homes and in the community.	To explore the validity, reliability and sensitivity to change of movement behaviors metrics from three activity trackers in free-living conditions when worn by older adults	Aged ≥60 years; able to walk independently without mobility aids; absence of substantial (self-reported) neurological and cognitive impairments	ND *	Approach included key measurement of properties of three commonly used activity trackers, in both controlled and free-living environments.	Results cannot be generalized to other populations, e.g., the older adult population with physical impairments that significantly affect ambulation. The use of research-grade monitor ActiGraph as a convergent measure as the ground-truth for free-living tasks was not ideal. Inability to define the level of user’s physical activity/fitness within the proprietary wearables could have affect the computed outputs.
Paul et al. [30]	Cross-sectional—validation study; all data collected at participants’ homes.	To investigate the criterion validity of Fitbit step counts compared to (i) visual count and (ii) ActiGraph accelerometer step counts. To investigate the accuracy of ActiGraph step counts compared to visual count in community-dwelling older people.	Aged over 60 years; lived at home; were regular (weekly) users of the internet via a computer or tablet device and left their house regularly (at least once per week) without physical assistance from another person.	Were housebound; had a cognitive impairment. diagnosis of dementia or a Memory Impairment score < 5; had insufficient English language skills to fully participate in the programme; had a progressive neurological condition or a medical condition precluding exercise; were currently participating in 150 min of moderate intensity physical activity per week and had undergone a fall risk assessment in the past year with subsequent adoption of recommendations.	ND *	ND *
Soltani et al. [42]	Cross-sectional—validation study; free-living measurements were carried out in participants’ homes and in the community.	To investigate the accuracy and precision of an algorithm designed to detect gait bouts in free-living conditions using a single three-dimensional accelerometer on a wrist, and on older adults.	ND *	ND *	ND *	ND *
Taylor et al. [33]	Cross-sectional—data collected at retirement village and participants’ homes.	To evaluate the validity of the DynaPort MoveMonitor system for detection of gait and postures (sitting, lying, standing) in people aged >80 years, using video observation as the reference measure.	Aged >80 years; were living in either independent or in long-term care (nursing home) facilities at the retirement village; able to transfer and walk independently with or without a walking aid.	ND *	ND *	The use of scripted test protocol could have increased error when testing short activity bouts. In real life, transitions are less common, and the duration of activity bouts, especially inactivity, is longer. Therefore, the unscripted test protocol might better reflect the accelerometer’s validity for everyday activity recognition than the short activity bouts used in the scripted test protocol.
Yamada et al. [43]	Cross-sectional—data collected at participants’ homes.	To examine the validity of a triaxial accelerometer in estimating total energy expenditure and physical activity levels in older adults with the doubly labelled water method.	Aged >64; weight is stable.	On medications known to affect weight, kidney function, or metabolism	ND *	Participants were recruited from ongoing health and physical function check-up cohort and institutionalized people. The participants of the annual check-up tend to have high motivation to be active to prevent decrease in physical function. Small sample size for generalisability. Selection bias may have occurred.

* ND—not described. ^1^ Bourke et al. [44,45]. This was based on 20 subjects who included obese older adults. Data for the 16 involved in the free-living studies alone not provided. ^2^—moderate-vigorous physical activity.

**Table 5 sensors-23-07615-t005:** Clinimetric properties and methods of studies included in the systematic review.

Study	Brand of PA Instrument	Testing Protocol	PA Metrics	Definitions of PA Metrics	Reliability	Validity	Responsiveness
Awais et al. [37]	uSense (on chest, L5 and thigh); Shimmer3 (on non-dominant wrist)	To use deep learning approach to classify physical activity; and to compare the performance of the deep learning approach with classical machine learning approach. Algorithms tested were PAC-LSTM, PAC-All-Feat, PAC-CFS, PAC-FCBF and PAC-ReliefF ^1^.	Participants performed unsupervised activities of daily living but were advised to include a set of defined activities ^2^ (see Bourke et al. for more details). All activities were performed in their own homes.	Sitting: When the person’s buttock is on the seat of the chair and their trunk is in a continuous relatively upright posture. Lying: The person’s body, including trunk and thigh are in a relatively horizontal posture supported along the length of the body. Standing: The person is in an upright posture with both feet supporting the person’s body weight, with no feet movement. Walking: Locomotion towards a destination with 1 stride or more, (minimum: 1 step with both feet, finishing where 1 foot is placed beside the other foot). ^2^	Inter-rater reliability ^2^ was calculated from five raters annotating one randomly selected video. Cohen’s Kappa was 90.05%.	F-scores (in %) were computed for sitting, lying, standing and walking activities. Overall F-score (mean of all classes) was used to compare between algorithms. PAC-LSTM: walking: 94.48%; sitting: 99.90%; standing: 96.09%; lying: 98.46%; Overall: 97.23% PAC-All-Feat: walking: 92.65%; sitting: 99.81%; standing: 95.48%; lying: 89.39%; Overall: 94.33% PAC-CFS: walking: 93.32%; sitting: 99.68%; standing: 95.29%; lying: 84.72%; Overall: 93.25% PAC-FCBF: walking: 86.91%; sitting: 99.69%; standing: 91.58%; lying: 86.49%; Overall: 91.17% PAC-ReliefF: walking: 93.48%; sitting: 99.95%; standing: 95.41%; lying: 98.46%; Overall: 96.83%	ND *
Brand et al. [38]	Garmin (Garmin International, Olathe, KS, US), AX3 (Axivity, UK) (as reference)	Participants wore Garmin on their nondominant wrist and AX3 on L5, for up to 10 days.	Gait bouts based on 6 s windows; Amount of daily walking.	ND *	ND *	The new algorithm had 76.2% accuracy, 29.9% precision, 67.6% sensitivity and 78.1% specificity in detecting gait bouts. The Pearson’s correlation coefficient for amount of daily walking was 0.84.	ND *
Briggs et al. [31]	Garmin Vivosmart HR (Garmin International, Olathe, KS, US), ActiGraph GT3X+ (Actigraph, Pensacola, FL, US) (as reference)	Participants wore Garmin and ActiGraph on their nondominant wrist. An additional ActiGraph was worn on the hip. Participants were instructed to wear all activity devices continuously, including sleep, except for water activities, for a minimum of 48 consecutive hours while not participating in any structured Gerofit exercise.	Daily average step counts and duration (in minutes) of MVPA.	ND *	ND *	Intraclass correlation (95% CI) between Garmin and ActiGraph (hip) was: (a) for daily step count: 0.94 (0.88, 0.97) (b) for MVPA (>2020 counts/min): 0.16 (−0.40, 0.55) (c) for MVPA (>1013 counts/min): 0.35 (−0.32, 0.70) (d) for MVPA (>1924 counts/min): 0.38 (−0.31, 0.71) BA plots revealed that Garmin overestimated MVPA compared with the hip worn ActiGraph. However, the difference was small using the lower, age specific, MVPA cut-off (see above)	ND *
Burton et al. [39]	Flex (FitBit, San Francisco, CA, US), ChargeHR (FitBit, San Francisco, CA, US), GENEactiv(ActivInsights Ltd., UK) (as reference for free-living)	Participants wore a randomly allocated Flex or ChargeHR fitness tracker and an accelerometer to wear for 14 days (including sleeping). The only exception was to remove when in water (e.g., shower or swimming).	Number of steps (for lab and free-living); distance walked (for lab only), sleep (for free-living) from Flex and ChargeHR. Total PA (in mins) (for free-living), MPVA (for free-living) and sleep (for free-living) from GENEactiv.	ND *	ND *	Construct validity—14-day free-living: strength of agreement (Spearman Rho’s) for steps (fitness tracker) and MPVA(GENEactiv) was 0.70 (−0.10, 0.96) (Flex—0.68, ChargeHR—0.72), steps (fitness tracker) and Total PA (GENEactiv) was 0.54 (−0.12, 0.90).	ND *
Chigateri et al. [32]	uSense	Scripted tasks—Two sets of TUG ^3^ (at usual gait speed), sit-to-stand and stand-to-sit transfers using a chair with arms and without arms. Unscripted tasks—activities that reflected what the participants normally do during the day. Both scripted and unscripted tasks were videoed and coded as sit to stand, stand to sit, sit to lie, lie to sit; lying; standing or active standing; shuffling; and walking.	Duration of tasks identified in “Testing Protocol” categorised as “walking” and “non-walking”.	Shuffling was defined as *“where there was forward ambulation but not clear strides”* Shuffling and walking were grouped as “walking”. Sit to stand, stand to sit, sit to lie, lie to sit; lying; standing or active standing—“non-walking”.	ND *	95% limits of agreement between the mean video time and the algorithm time categorization during scripted activity was 1.47 s (−4.69 to 7.63) for walking and for −1.47 s (−7.63 to 4.69) non-walking. 95% limits of agreement between the mean video time and the algorithm time categorization during scripted activity was 26.5 s (18.8 to 71.6) for walking and −26.5 s (−71.6 to −18.8) for non-walking. Algorithm identified walking episodes for unscripted and scripted activities with 92.8% and 95.1%, respectively. For scripted activity, 97.2% and 91.4% agreement were achieved between the video and the algorithm for non-walking and walking activity, respectively. For unscripted activity, 92.2% and 88.7% agreement were achieved between the video and the algorithm for non-walking and walking activity, respectively.	ND *
Dijkstra et al. [40]	MiniMod (DynaPort)	Participants performed a fixed activity sequence including walking trajectories (1.4 m, 2.3 m, 4.5 m, taking a three-step stair) and postures (sitting, standing, lying), five times. Thereafter, they were allowed to move freely for 3 min with the only instruction that taking the stairs, sitting and lying had to be completed at least once. They also performed usual domestic activities (such as doing the dishes, watering plants, hanging up laundry or mowing the lawn). During all activities, participants were video recorded. Start and end of each activity was scored by an observer from the video analysis. Inter-rater reliability was determined for the fixed sequence task by two raters for 10 participants.	Mean activity duration of walking, sitting, standing and lying,	Walking was determined, starting from the heel-off for the initial step until ending with full floor contact of the foot making the last step ^4^, and the number of steps taken 2 or more. Persons were considered to be sitting when their upper body was upright and at a 90° angle to the legs. Standing was determined when the participant was in an upright position with no or a small displacement, but no distinctive steps, of the feet. Lying was defined as the person being in a horizontal position and either the side or the back of the body contacting the bed.	Inter-rater reliability was calculated from two independent raters. Intraclass correlation coefficient (ICC) for the duration of walking, sitting, standing and lying were 0.95, 0.78, 0.99 and 0.98, respectively.	Agreement per participant ranged between 68.3 and 85.9% (mean = 79.8%; SD = 6.9). Sensitivity for duration of walking, sitting, standing and lying were 93.5%, 83.2%, 80.1% and 98.7%, respectively. Specificity for duration of walking, sitting, standing and lying were 71.8%, 78.7%, 77.7% and 77.6%, respectively. Positive predictive value for duration of walking, sitting, standing and lying were 80.7%, 76.8%, 80.2% and 64.6%, respectively.	ND *
Farina et al. [29]	Fitbit Charge HR (FitBit, San Francisco, CA, US), Misfit Shine (Misfit Wearables), ActiGraph GT3X+ (Actigraph, Pensacola, FL, US) (as reference), NL2000i (New-Lifestyles Inc, Lee’s Summit, MO, US) (as reference).	Participants wore all 5 devices: on dominant wrist—Fitbit Charge HR and Misfit Shine, waist (dominant side)—Misfit Shine, ActiGraph GT3X+ and NL2000i (not described further).	step count, steps/day	ND *	ND *	Fitbit Charge HR (wrist)—ICC:0.86 (0.68 to 0.94) with ActiGraph GT3X+; 0.85 (0.65 to 0.94) with NL2000i. Misfit Shine (wrist) ICC:0.86 (0.67 to 0.94) with ActiGraph GT3X+; 0.83 (0.59 to 0.93) with NL2000i. Misfit Shine (waist) ICC:0.96 (0.91 to 0.99) with ActiGraph GT3X+; 0.91 (0.79 to 0.97) with NL2000i. Bland–Altman plots—compared with and both references “*near perfect*” agreement for Misfit (Waist), “*moderately wide*” agreement for Misfit (Waist), “*very wide*” agreement for Fitbit (Wrist).	ND *
Geraedts et al. [41]	Philips (Philips Research, Eindhoven, Netherlands)	Standardized movement protocol: Participants performed TUG (slow, normal, fast), Five Times Chair Rise, standing still, walking, lying and sitting with rest in between, if required. Free movement protocol: 30 min of self-chosen activities (e.g., vacuuming, reading, preparing tea or coffee, cleaning dishes and watering plants). All activities, in both protocols, were videoed, annotated and scored, namely for sitting, standing, walking and lying by three (two/three for watch video) independent raters. Participants wore the sensor over 1 week and provided feedback on comfort, weight, size and usability via questionnaire.	Total duration (in seconds) of Time-on-leg (ToL), sitting, standing, walking and lying.	ToL: the time spent actively on the legs, i.e., standing, shuffling around, walking ^5^ and transitions in between. Lying was defined when the person’s trunk was in a horizontal position with the back, stomach or side touching a horizontal underground without signs of further movement. Sitting was defined when the person’s trunk was in a vertical seated position without movement in the trunk. Standing was defined when the person was in an upright vertical position with no or only a small displacement, but no distinctive steps, of the feet.	Percentage of agreement was calculated for assessment of inter-rater reliability on the video annotation. ICC for free-living protocol was 0.91.	Overall agreement for non-frail participants: Standardized movement protocol: TOL—79.2%, sitting—72.9%, standing—75.9%, walking—93.3%, lying—96.9%. Free movement protocol: TOL—85.0%, sitting—84.6%, standing—70.7%, walking—86.2%, lying—99.5%. Overall agreement for frail participants: Standardized movement protocol: TOL—86.0%, sitting—78.8%, standing—83.0%, walking—92.6%, lying—97.5%. Free movement protocol: TOL—91.6%, sitting—85.1%, standing—77.4%, walking—90.9%, lying—99.9%.	ND *
Kastelic et al. [28]	Polar Vantage M (Polar Electro OY, Kempele, Finland); Garmin Vivoactive 4s (Garmin, Olathe, KS, US); Garmin Vivosport (Garmin, Olathe, KS, US); ActiGraph wGT3X-BT (Actigraph, Pensacola, FL, US) (as reference).	Participants wore a randomly assigned activity tracker on their non-dominant wrist, whilst wearing the ActiGraph on their waist over their dominant leg, for either six days (baseline protocol group) or 4 days (extended protocol group). ^6^	Daily step count, intensity minutes ^7^ (calculated), active calories burned ^7^ (calculated).	Sedentary behaviour cut-off was 0–99 cpm ^8^; Light activity was 100–2019 cpm; moderate activity was 2020–5998 cpm; vigorous activity was 5999 cpm and above.	Intraclass correlation coefficients (ICC2,1) for single-day daily step counts: Polar Vantage M—0.68 [0.43, 0.85]; Garmin Vivoactive 4s—0.70 [0.44, 0.88]; Garmin Vivosport—0.65 [0.39, 0.84]. Intraclass correlation coefficients (ICC2,1) for three-day average step counts: Polar Vantage M—0.82 [0.48, 0.94]; Garmin Vivoactive 4s—0.24 [−0.56, 0.81]; Garmin Vivosport—0.66 [0.13, 0.89]. Intraclass correlation coefficients (ICC2,1) for single-day daily active kcal: Polar Vantage M—0.80 [0.61, 0.91]; Garmin Vivoactive 4s—0.66 [0.38, 0.85]; Garmin Vivosport—0.48 [0.18, 0.74].	Agreement (ICC2,1) between Polar Vantage M and ActiGraph for steps: 0.37 (*p* = 0.001) Agreement (ICC2,1) between Garmin Vivosport and ActiGraph for steps: 0.98 (*p* = 0.000) Agreement (ICC2,1) between Garmin Vivoactive 4s and ActiGraph for steps: 0.95 (*p* = 0.000) Agreement (ICC2,1) between Polar Vantage M and ActiGraph for active kcal: 0.15 (*p* = 0.056) Agreement (ICC2,1) between Garmin Vivosport and ActiGraph for active kcal: 0.58 (*p* = 0.001) Agreement (ICC2,1) between Garmin Vivoactive 4s and ActiGraph for active kcal: 0.55 (*p* = 0.011)	Between subjects’ responsiveness over a single day activity for steps count (Guyatt’s responsiveness coefficient (GR)): Polar Vantage M—0.126; Garmin Vivosport—0.411; Garmin Vivoactive 4s—0.022 Between subjects’ responsiveness over 3-day activity for steps: Polar Vantage M—0.060; Garmin Vivosport—0.041; Garmin Vivoactive 4s—0.288
Kastelic et al. [28]					Intraclass correlation coefficients (ICC2,1) for three-day average active kcal: Polar Vantage M—0.86 [0.59, 0.96]; Garmin Vivoactive 4s—0.54 [−0.26, 0.90]; Garmin Vivosport—0.66 [0.13, 0.90]	Bland–Altman plots revealed that all devices overestimated step counts: Polar Vantage M (6719 ± 4168 steps), Garmin Vivosport (740 ± 1262 steps) and Vivoactive 4s (639 ± 796 steps) Minimal detectable change in step counts over a single day (steps): Polar Vantage M: 10,832 Garmin Vivosport: 9592 Garmin Vivoactive 4s: 7714 Minimal detectable change in step counts averaged over three valid days (steps): Polar Vantage M: 6178 Garmin Vivosport: 6987 Garmin Vivoactive 4s: 5864 Minimal detectable change of active calories over a single day (kcal): Polar Vantage M: 597 Garmin Vivosport: 572 Garmin Vivoactive 4s: 446 Minimal detectable change of active calories averaged over three valid days: Polar Vantage M: 368 Garmin Vivosport: 380 Garmin Vivoactive 4s: 289	Between subjects’ responsiveness over a single day active kcal: Polar Vantage M—0.232; Garmin Vivosport—0.261; Garmin Vivoactive 4s—0.073 Between subjects’ responsiveness over 3-day active kcal: Polar Vantage M—0.248; Garmin Vivosport—0.933; Garmin Vivoactive 4s—0.536
Paul et al. [30]	One (FitBit, San Francisco, CA, US); Zip (Fitbit, San Francisco, CA, US); ActiGraph GT3X+ (Actigraph, Pensacola, FL, US)	Participants performed a 2MWT ^9^ in the space available in their homes. Number of steps was visually counted by research physiotherapist using a hand-held stationery counter. Participants were instructed to stand still for 10 s prior to and after the 2 MWT, and the start and finish times of the 2 MWT were recorded. Participants also wore the Fitbit simultaneously with the ActiGraph during waking hours (except for water sports or bathing) for a 7-day period. Completed a physical activity log for weeklong period which data was checked against for inconsistencies and erroneous data.	steps/day	Fitbit—step counts per day estimated based on proprietary algorithm; ActiGraph—step counts in 60 s epochs and Freesdon Adult (1998) ^10^ equation.	ND *	2 MWT—Fitbit versus Observer (ICC2,1 = 0.88, 95% CI 0.76 to 0.94), Fitbit and ActiGraph (ICC2,1 = 0.66, 95% CI 0.41 to 0.82), ActiGraph versus Observer (ICC2,1 = 0.60, 95% CI 0.33 to 0.79). Average steps/day—Fitbit versus ActiGraph (ICC2,1 = 0.94, 95% CI 0.88 to 0.97) but Fitbit had 716.7 more steps/day (95% CI 318.2 to 1115.1). Bland–Altman plots revealed a bias by the ActiGraph for people who took fewer steps during the 2 MWT. Bland–Altman plot revealed no systematic bias in averaged daily step counts between the Fitbit tracker and ActiGraph accelerometer. There was less percentage agreement between the Fitbit and ActiGraph for average daily steps with 34–66% of participants having Fitbit scores within 5–15% of ActiGraph scores.	ND *
Soltani et al. [42]	GENEactiv Original (ActivInsights Ltd., UK), ActiGraph GT9X (ActiGraph, Pensacola, FL, US) (as reference)	Participants wore GENEactiv Original on the wrist, whilst wearing the ActiGraph GT9X on the shank, for a continuous 12 h in real-world situations.	Total duration of gait bouts.	A walking period is defined as an interval with at least 3 successive steps. ^11^	ND *	Leave-one subject-out cross validation for total duration of gait bouts resulted in the following: sensitivity was 87.1% [72.6, 91.8], specificity was 96.7% [95.5, 97.6], accuracy was 95.2% [94.1, 96.7], precision was 71.8% [56.4, 76.3] and F1-score was 74.9% [63.6, 83.6].	ND *
Taylor et al. [33]	MoveMonitor (DynaPort)	Scripted ^12^: (in retirement village) performed a TUG at their usual walking pace; a 4.5 m walk and lie down on a bed for 30 s; rise from lying and stand for 30 s; and walk back to the chair and sit down. The sequence was performed twice, taking 4 to 6 min in total to complete.	Total duration (in seconds) and mean duration (in seconds) of sitting, standing, locomotion and lying.	Based on Dijkstra et al. [40]	Inter-rater reliability was calculated from two independent raters. The ICC (average measures) for sitting, standing, locomotion, and lying were 0.99, 0.98, 0.94, and 0.99, respectively.	Median percentage of error: sitting, −22.3% (IQR, −62.8% to 10.7%); standing, 24.7% (IQR, −7.3% to 39.6%); locomotion, 0.2% (IQR, −4.3% to 14.0%); and lying, 0.3% (IQR, −4.2% to 21.4%). ^13^ Sensitivity (Unscripted): sitting—94.5 (91.1–97.0); standing—38.6 (10.7–86.2); locomotion—92.2 (84.2–97.3); lying—100 (97.9–100) Specificity (Unscripted): sitting—81.4 (78.1–98.8); standing—96.8 (91.9–98.4); locomotion—97.1 (96.3–99.1); lying—99.2 (99.0–100)	ND *
Taylor et al. [33]		Unscripted (at home): move about freely; walking, sitting, standing, and lying down in a random sequence for between 5 and 9 min within their own home environment. Instruction was limited to ensure that activities were performed in their normal manner. Both scripted and unscripted tasks were recorded using a handheld digital video camera that was synchronized with the accelerometer.			ND *	Overall agreement (Unscripted): sitting—85.2 (78.7–91.5); standing—56.1 (34.8–81.2); locomotion—89.9 (80.8–94.7); lying—98.0 (73.8–100). When misclassified activities were analysed further, standing was found to be incorrectly classified as sitting for 28.1% of the scripted and 45.6% of the unscripted total standing time. Sitting was misclassified as standing for 14.9% and 5.3% of the total sitting time for the scripted and unscripted tasks, respectively.	ND *
Yamada et al. [43]	Actimarker (Panasonic, Osaka, Japan) Lifecorder (Kenz, Suzuken, Japan)	Basal metabolic rate (BMR) obtained in the laboratory over 12 wakeful hours. Total energy expenditure (TEE) was measured using doubly labelled water (DLW) method over 14 days (iii) Physical activity levels (via Actimarker) and step counts (via Lifecorder) were obtained based on “usual daily activities” over 14 days.	TEE measured via DLW method over 14 days based on step count and steps/day.	Based on IAEA ^14^	ND *	The 24 h average MET of ACCTRI was significantly correlated with PAL of DLW but significantly underestimated it (*p* < 0.001). TEE of the ACCTRI systematically underestimated actual TEE (−14.2 ± 11.6%). Correlation between 24 h average MET of ACCTRI and PAL of DLW was R^2^ = 0.475, *p* < 0.001. Correlation between daily step counts and PAL of DLW was R^2^ = 0.248, *p* < 0.001.	ND *

* ND—not described. ^1^ Refer to Awais et al. [37] for more details on these algorithms. ^2^ From Bourke et al. [45]. See Bourke et al. [44] for a complete list of activities recommended included sitting, lying, preparing food or drink while standing and setting up the table. See also Awais et al. [47]. ^3^ TUG—Timed Up and Go assessment. ^4^ A step was defined as a forward displacement of the foot together with a forward displacement of the trunk. ^5^ Walking was defined when the person was moving the feet forward in a walking pattern with the trunk in a forward displacement, from when the heel of the foot cleared the ground for the initial step until the foot of the closing step made complete contact with the floor, with a minimum of 2 steps. ^6^ Only data from the extended protocol group is discussed. ^7^ Only step counts and active calories burned are reported here as common metrics between the three accelerometers and the reference, ^8^ count per minute (cpm). ^9^ 2 MWT—2 m walk test ^10^ [48]. ^11^ Based on Najafi et al. [49]. ^12^ Only results of unscripted data is further discussed in this paper. ^13^ These are based on combined durations of scripted and unscripted activities. Separate data on scripted and unscripted activities were not provided ^14^ [50].

**Table 6 sensors-23-07615-t006:** Summary of clinimetric properties of PA measures in real-world conditions.

Clinimetric Property	Measures	Range
*Inter-rater reliability* (ICC ^1^)	walking duration	0.94–0.95
	lying duration	0.98–0.99
	sitting duration	0.78–0.99
	standing duration	0.98–0.99
*Relative reliability* (ICC)	step counts	0.24–0.82
	active calories	0.48–0.86
*Absolute reliability* (minimal detectable change)	step counts ^2^	5864–10,832
	active calories ^3^	289–597
*Responsiveness* (Guyatt’s responsiveness coefficient)	step counts	0.02–0.41
	active calories	0.07–0.93
*Criterion validity*	step counts (ICC)	0.83–0.98
	walking duration (%)	63.6–94.5
	lying duration (%)	35.6–100.0
	sitting duration (%)	79.2–100.0
	standing duration (%)	38.6–96.1
*Construct validity* (Spearman’s Rho)	step counts and MVPA	0.68–0.72

^1^ intraclass correlation coefficient; ^2^ in counts; ^3^ in kilocalories.

## Data Availability

Not applicable.

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
