# Peer review of "Physical Activity in Community-Dwelling Older Adults: Which Real-World Accelerometry Measures Are Robust? A Systematic Review"

_sensors, 2023, doi:10.3390/s23177615_

Round 1
Reviewer 1 Report
1. Question mark in the title does not make sense.
2. Table 2 is not make understable.
3. In the table-3 "basic study characteristic", whose characteristics? its is not understandable.
4. some subtitles and highlights are very short. kindly make a clear view on it.
5. Abstract and conclusion should be more accurate.
it need to improve bit.
Author Response
Dear Reviewer,
We would like to thank the reviewers for their valuable time and comments which have helped us to improve the manuscript. Please find our responses in the attachment.
Thank you.

Reviewer 2 Report
Dear authors,
I read your manuscript with interest. I appreciate your work, mainly for the large number of references analyzed and then selected (using 6 clearly defined inclusion and exclusion criteria) from the 8 databases consulted. You have investigated the utility and limitations of using wearable sensors including accelerometers to measure various physical activity variables in the elderly. I can offer you some minor suggestions to improve the initial draft of the peer-reviewed manuscript:
1. The ICC results for the parameters included in the abstract would be useful to summarize and present in a table in the text of your article. In this way it would be easier to understand.
2. In your abstract it would be helpful to include a defining conclusion for your research.
3. Line 449: 3.12. Acceptability and adherence of accelerometers. I think it's 3.13, numbering 3.12. it is already at line 440.
4. Tables 4, 5 and 6 present very important data synthesized from the 13 scientific articles analyzed. However, they include more than half of the number of pages of the article. Perhaps you can format them (if this is possible) to reduce the final size of the manuscript.
5. You mentioned 3 distinct research directions at the end of the introduction. It would be useful to adapt/correlate the conclusions with these formulated ideas.
Author Response

(The authors gave the same response as above.)

Reviewer 3 Report
The paper is well written. The subject is within the scope of the journal, and the research objective is well stated. The presentation is very good.
However, it doesn't adequately emphasize the issues related to the duration of experiments that are conducted in many studies. This study (https://doi.org/10.3389/fphys.2020.00090) provides data highlighting a significant difference between offline and online data processing concerning the reliability of measurements for diagnostic purposes. Specifically, long off-laboratory offline processing is more reliable than in-lab online measurements. At the same time, many studies propose the use of platforms for long-term activity (e.g., https://doi.org/10.1109/TBCAS.2022.3173586, a document that could be cited in the text).
I believe that this aspect is crucial and should not be overlooked. At the very least, it should be addressed in the discussion.
1. Mainly the English is good and there are only a few typos. However, the paper should be carefully rechecked.
Author Response

(The authors gave the same response as above.)

Reviewer 4 Report
This is a systematic review paper on a subject of high interest.
Physical Activity in Community-Dwelling Older Adults: Which Real-World Accelerometry Measures are Robust?
The text is well written and in good English.
The methodology followed seems to be the most appropriate and the results obtained support the conclusions.
Small details to improve the text:
In this work, two types of references coexist, number between straight parentheses and author's name with the year of publication between curved parentheses. I think that a single form should be adopted.
In the abstract
what ICCs represent, is it “Intraclass correlation coefficient?”, must be explained in full at first use.
Line 48
[adapted from 12] replace with (adapted from [12])
Line 270
DLW method replace with Doubly Labeled Water (DLW) although it appears on 435 must be explained at first use.
In my opinion the work deserves publication with minor changes.
Author Response

(The authors gave the same response as above.)

Round 2
Reviewer 1 Report
The paper is nicely written, the only concern is add few papers on the given topic.
1. Satpathy, S., Das, S., & Bhattacharyya, B. K. (2020). How and where to use super-capacitors effectively, an integration of review of past and new characterization works on super-capacitors. Journal of Energy Storage, 27, 101044.
2. Sengupta, A. S., Satpathy, S., Mohanty, S. P., Baral, D., & Bhattacharyya, B. K. (2018). Supercapacitors outperform conventional batteries [energy and security]. IEEE Consumer Electronics Magazine, 7(5), 50-53.
3. Satpathy, S., Dhar, M., & Bhattacharyya, B. K. (2020). Why supercapacitor follows complex time-dependent power law (t∝) and does not obey normal exponential (e− t (RC)) rule?. Journal of Energy Storage, 31, 101606.
4. Satpathy, S., Padthe, A., Prakash, M., Trivedi, M. C., Goyal, V., & Bhattacharyya, B. K. (2022). Method for measuring supercapacitor’s fundamental inherent parameters using its own self-discharge behavior: A new steps towards sustainable energy. Sustainable Energy Technologies and Assessments, 53, 102760.
Reviewer 3 Report
Authors have properly enriched their work, by addressing each comment in a suitable way. The paper turns out to be notably improved.
Minor editing of English language required